# Soft Value Iteration Networks for Planetary Rover Path Planning

## Abstract

Value iteration networks are an approximation of the value iteration (VI) algorithm implemented with convolutional neural networks to make VI fully differentiable. In this work, we study these networks in the context of robot motion planning, with a focus on applications to planetary rovers. The key challenging task in learning-based motion planning is to learn a transformation from terrain observations to a suitable navigation reward function. In order to deal with complex terrain observations and policy learning, we propose a value iteration recurrence, referred to as the soft value iteration network (SVIN). SVIN is designed to produce more effective training gradients through the value iteration network. It relies on a *soft* policy model, where the policy is represented with a probability distribution over all possible actions, rather than a deterministic policy that returns only the best action. We demonstrate the effectiveness of the proposed method in robot motion planning scenarios. In particular, we study the application of SVIN to very challenging problems in planetary rover navigation and present early training results on data gathered by the Curiosity rover that is currently operating on Mars.

## 1 Introduction

Value iteration networks (VIN) are an approximation of the value iteration algorithm (Bellman, 1957; Bertsekas, 1995) that were originally proposed by Tamar et al. (2016) as a way of incorporating a differentiable planning component into a reinforcement learning architecture. In this work, we apply the technique in a fully supervised, imitation learning approach for robot path planning problems. The architecture has two main neural network components, the VIN itself which is an unrolling of the value iteration recurrence to a fixed number of iterations, and the reward network which transforms terrain and goal data into a reward map that feeds into the VIN. An important feature of this approach is that the learned component of the network exists entirely within the reward network, the output of which must be a well behaved reward function for our planning problem, making human interpretation of the planning results relatively easy. In this work, we restrict ourselves to 2D path planning problems on grids that have only local neighbors. This is a natural constraint of using convolutional neural networks for implementing the value iteration algorithm, although one could imagine convolutional VINs at higher dimensionality.

Although the use of a reward function in value iteration is quite intuitive, writing down how to calculate a reward function to produce the desired planning results is deceptively difficult, as has been observed by researchers in the past (Bagnell et al., 2010). Part of the difficulty comes from the need to keep the relative costs and rewards of different types of terrain in balance with each other.

### 1.1 Planetary Rover Path Planning

Planetary rovers, such as those currently being operated on Mars, face difficult navigation problems in both short and long range planning (Gaines et al., 2016). These problems are made more difficult by tight bandwidth constraints and time lag to communicate with Earth that operationally limit the rovers to only making one round trip communication with Earth per sol (a sol is a Martian day, about 24 hours 40 minutes). Existing rover driving techniques rely mostly on imagery taken on the surface to plan actions, however those plans can only go as far as the rover can see.

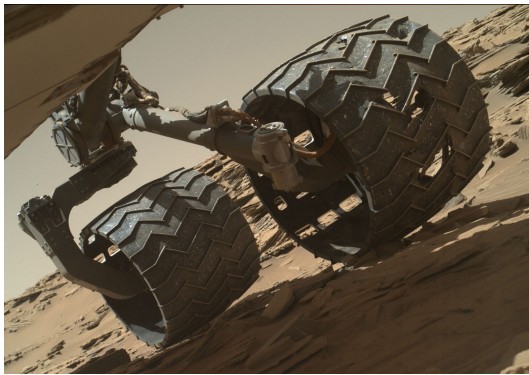

Figure 1: MSL image showing damage to the wheels from small rocks.

Rovers on the ground are naturally limited in their ability to see by variations in terrain and obstructions, as well as by limitations of the cameras. Orbital imagery can be provided at effectively unlimited distance, but arrives at lower resolution than images on the ground. Although orbital imagery can be processed to obtain certain characteristics of the terrain, such as coarse elevation, or in some cases estimates of terrain type, relating these characteristics to a cost function relevant to navigation is nontrivial. In particular, many dangers to the Mars Science Laboratory (MSL) "Curiosity" rover are too small to be seen directly in orbital imagery (Figure 1), but it might be possible to find terrain or mineral patterns that are associated with dangerous terrain.

Autonomous driving techniques also exist, though they lack the ability to do high level planning to choose paths that are advantageous for longer term navigation. In this work, we aim to use orbital imagery to produce longer range rover plans that will enable future rovers to reduce their reliance on communication to Earth for navigation.

We observe that this problem possesses two important properties: we can formulate useful algorithms for long range planning if a suitable cost function is available and short range planning techniques exist (based on surface imagery) that are both sophisticated and reliable.

In this work, we present an imitation learning architecture based on value iteration networks that implicitly learns to solve navigation planning problems using training data derived from historical rover driving data. These navigation functions only depend on orbital data to produce a plan, and are thus useful at ranges beyond what is visible from the ground. Because the imitation learning architecture we use is structured around value iteration, it implicitly produce useful cost functions for navigation. We propose an architecture that allows this planning product to be integrated with existing human expert-designed local planners to maintain guarantees in safety critical systems.

## 2  RELATED WORK

Deep reinforcement learning (RL) techniques have been frequently applied in settings where we wish to model an action policy as a function of robot state for both discrete and continuous actions spaces (Mnih et al., 2015; Lillicrap et al., 2015; Schulman et al., 2015).

Value iteration networks were first explored by Tamar et al. (2016) in the context of reinforcement learning. We have taken the value iteration module from their work, but instead of allowing it to have abstract spatial meaning and pass through a final policy network, we bind it tightly to the map of our terrain and the available state transitions. Other researchers have also looked at ways of using neural networks to model dynamic programming and planning techniques (Ilin et al., 2007; Silver et al., 2017).

Imitation learning for navigation has been studied by other groups as well (Silver et al., 2010), for example the LEARCH technique proposed by Ratliff et al. (2009) has a very similar functional input/output design, although it relies on substantially different computational tools. Imitation learning has also been done with DNN based policies and visual state information (Giusti et al., 2016).

## 3 PRELIMINARIES

We use the problem formulation of a Markov Decision Process $M = (S, A, P, R, \rho)$ where $\rho(s)$ defines the distribution over initial states, and a robot in state $s \in S$ must choose an action $a \in A$, which will give reward $r = R(s, a)$ and proceed to state $s'$ with probability $P(s'|s, a)$. We say that a policy $\pi(a|s)$ will define a policy distribution over actions conditioned on states, and that $\tau^\pi(s_0, a_0, s_1, a_1, ...)$ is the distribution over trajectories created recursively when $s_0 \sim \rho(s)$, $a_0 \sim \pi(a|s_0)$, and $s_1 \sim P(s_1|s_0, a_0)$.

We then define the value function $V$ conditioned on policy $\pi$ as

$$V^\pi(s) = \mathop{\mathbb{E}}_{s,a \sim \tau} \left[ \sum_{t=0}^{\infty} \gamma^t R(s_t, a_t) \right] \tag{1}$$

for some discount factor $\gamma \in [0, 1]$. This can be rewritten recursively as

$$V^\pi(s) = \mathop{\mathbb{E}}_{a \sim \pi(a|s)} \left[ R(s, a) + \gamma \sum_{s'} P(s'|s, a) V^\pi(s') \right] \tag{2}$$

We further define $Q^\pi$ and rewrite the value function:

$$Q^\pi(s, a) = R(s, a) + \gamma \sum_{s'} P(s'|s, a) V^\pi(s') \tag{3}$$

$$V^\pi(s) = \mathop{\mathbb{E}}_{a \sim \pi(a|s)} [Q^\pi(s, a)] \tag{4}$$

We define the optimal value function $V^*(s) = \max_\pi V^\pi(s)$ and a policy $\pi^*$ as the optimal policy if $V^{\pi^*} = V^*$. The value iteration algorithm can be used to find $V^*$ through the following iterative process:

$$Q_n(s, a) = R(s, a) + \gamma \sum_{s'} P(s'|s, a) V_n(s') \tag{5}$$

$$V_{n+1}(s) = \max_a Q_n(s, a) \tag{6}$$

It is well known that as $n \to \infty$, $V_n \to V^*$, and an optimal policy $\pi$ can be inferred as $\pi^*(a|s) = 1_{\text{argmax}_a Q_\infty(s,a)}(a)$.

## 4 ALGORITHM

We adopt elements of the Value Iteration Network (VIN) architecture proposed by Tamar et al. (2016), however we change the output stage so that we now require the value iteration module to operate directly on the state space of our planning problem (rather than an inferred state space as in Tamar et al. (2016)). Although this restricts what can be done in the learning process, by forcing a traditional interpretation on the value map we can use it with any control policy that is designed to incorporate that data. In particular this includes the expert designed local control algorithms that provide the safety guarantee necessary for operating high value planetary rovers.

### 4.1 ARCHITECTURE

The data flow of our algorithm is shown in Figure 2. We start by stacking visual map data along with a one-hot goal map and any other data layers that are relevant to the application. These pass through the network $f_R$ to produce the reward map that feeds into the value iteration module. Also feeding into the value iteration module are the state transition and reward kernels, shown here at $f_P$.

The output from the VI module is used differently depending on whether we are currently training the algorithm or deploying it on an operational system. During deployment the value map from the VI module can be fed to an expert designed local planner and used in combination with high resolution local information to make planning decisions for a long range goal. During training the output from the VI module is fed to an action selection function to compare those results against actions chosen in the training data.

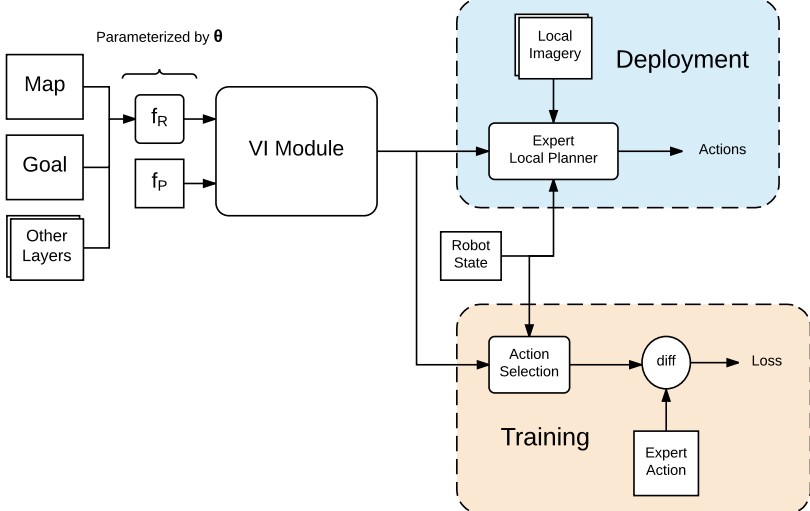

Figure 2: Information flow diagram for our architecture.

## 4.2 TRAINING

In general this architecture can have traninable parameters in both $f_R$ and $f_P$, however in our problems we fix the transition and reward kernels $f_P$ and only train parameters in the network $f_R$. These parameters can be trained with any normal supervised learning technique. We define our loss function as

$$L_\theta = - \sum_{s \in S_y} \sum_{a \in A} y(s,a) \log \frac{\exp Q_\theta(s,a)}{\sum_{i \in A} \exp Q_\theta(s,i)} \tag{7}$$

where $S_y$ is the set of states on the training path and $y$ is an indicator function for the action taken at each state on the training path. This may be recognized as a common softmax cross-entropy loss function.

## 4.3 PLAN EXECUTION

The network is provided (as input) an orbital image of the relevant terrain, some data products derived from that imagery such as stereo elevation and surface roughness estimates, and a goal position. With a forward pass through the network we calculate the value estimates based on those costs and send the value function of the rover for execution. Within the radius of it's sight, the rover is able to use expert algorithms for navigation, but can combine calculated path costs with value estimates at the perimeter of it's sight to choose the appropriate plan. Replanning can then happen as frequently as is allowed by on board resources.

## 4.4 VALUE ITERATION MODULE

The value iteration module uses the value iteration iterative process defined in equations 5 & 6 to perform an approximation of value iteration for a fixed number of iterations $k$. The approximate nature of this module derives from the need to choose the fixed number of iterations $k$ a priori, instead of iterating to convergence as required by the traditional value iteration algorithm. A representation of the architecture of the value iteration module is shown in Figure 3.

The two inputs $f_R$ and $f_P$ provide the reward map and convolutional kernel respectively. The reward map is stacked with the value map from the previous iteration and then convolved with $f_P$ to produce a map of Q values. The Q channels must then be collapsed into the next value map. Strict adherence

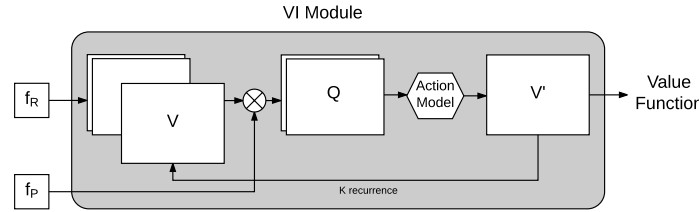

Figure 3: Value Iteration Module

to the value iteration algorithm requires that this be a max pooling operation, however, in the next section we propose an alternative approach.

### 4.5 SOFT ACTION POLICY

The traditional formulation of the value iteration algorithm requires that updates be done using the optimal action policy of always choosing the action of highest Q value as in Eq. 6 above. This is a theoretically well justified choice for a planning algorithm. However, in value iteration networks we have an additional objective to provide an effective gradient through the algorithm. If we assume a reward function $R_\theta$ parameterized by $\theta$, we can calculate the gradient of the value function with respect to $\theta$ after $k$ iterations as:

$$\nabla_\theta V_k(s) = \nabla_\theta R_\theta(s, a^*) + \gamma \sum_{s'} P(s'|s, a^*) \nabla_\theta V_{k-1}(s') \tag{8}$$

where, $a^*$ is the optimal action selected by the max $Q$ value in iteration $k - 1$. Assuming a deterministic state transition model $P$, this equation can be further simplified and expanded as:

$$\nabla_\theta V_k(s) = \nabla_\theta R_\theta(s, a^*) + \gamma \nabla_\theta V_{k-1}(s') \tag{9}$$

$$\nabla_\theta V_k(s) = \nabla_\theta R_\theta(s, a^*) + \gamma \nabla_\theta R_\theta(s', a'^*) + \gamma^2 \nabla_\theta V_{k-2}(s'') \tag{10}$$

The key observation from Eq. 10 is that the gradient through the value function will only incorporate information about states that are reached by the best actions under the current reward function. In this work, we propose a modification to the value iteration algorithm that leads to more effective gradient values, which enhances the network training, particularly in the early training stages. Instead of using the value update from Eq. 6, we propose the following:

$$V_{n+1}(s) = \sum_a Q_n(s, a) \frac{\exp(Q_n(s, a))}{\sum_{i \in A} \exp(Q_n(s, i))} \tag{11}$$

This formulation can be interpreted as performing value iteration under the assumption of a probabilistic action policy rather than an optimal action policy. In the traditional formulation with deterministic state transitions, the reward gradient cannot carry information about action selections in sub-optimal successor states. However, we can see that if we attempt to take the gradient of Eq. 11 w.r.t. $\theta$, we will not be able to remove the sum over actions and corresponding successor states. As such all possible future states and actions will (recursively) participate in the gradient.

Our experiments show that although the network can be trained in either model, using this 'soft' action model produces higher accuracy imitation learning results (Figure 5).

## 5 EXPERIMENTS

In this section, we demonstrate the performance of the proposed imitation learning framework. We are testing our algorithm with two datasets. The main dataset is a synthetic one that shows the performance and functionality of the proposed algorithm. We will compare the proposed method

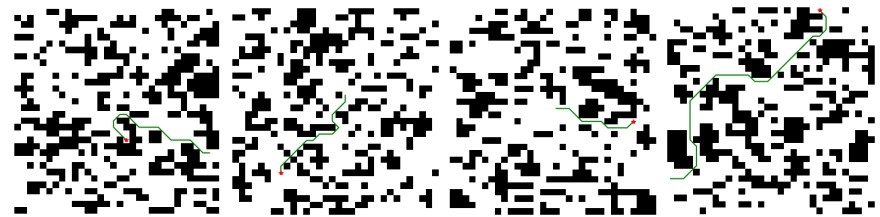

Figure 4: Four example maps and paths from our rock-world dataset. The star shows the goal position with one example path to that goal.

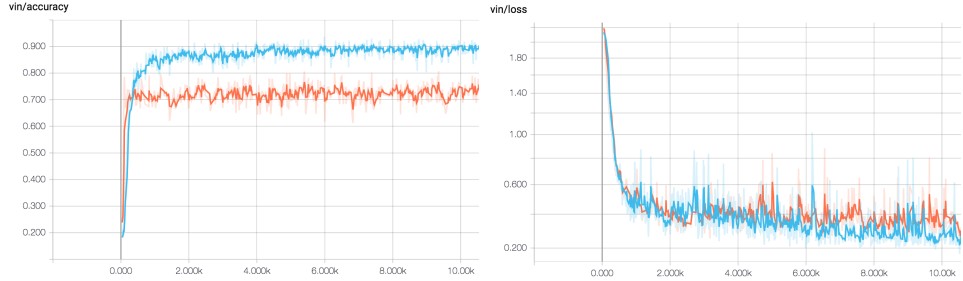

Figure 5: Training curves show accuracy and loss on the test set against gradient step on the rock-world dataset. The model in orange uses a standard hard action model, the light blue uses our proposed soft action model.

with other related work on this dataset. The second dataset corresponds to the Curiosity Mars rover data, gathered during years of driving the rover on Mars.

We track the performance of our models using two metrics, loss and accuracy. Loss refers to the standard softmax cross-entropy loss function described earlier in the paper. Loss is the function that is being optimized via gradient descent. Accuracy refers to the number of steps along the mission when optimal actions predicted by our network match the actions selected by the path data. The accuracy metric is not differentiable and hence cannot be optimized directly, however it is a better proxy for the useful performance of the network than the loss metric. Therefore, we primarily track progress in the accuracy value during training.

## 5.1 ROCK WORLD

Cumulative fractional area (CFA) of rocks (Golombek et al., 2012) is a rock distribution model on Mars surface. Inspired by the CFA model, we create a set of rock world map. Then, we discretize the rock worlds to 32x32 binary obstacle/free traversibility maps. For each map we randomly choose start and goal positions and compute the shortest path. The full dataset consists of 1000 maps with approximately 50 paths per map. The number of paths per map is approximate because occasionally a start a goal position are chosen for which there is no valid path in the map. This is the main dataset on which we evaluate and compare the performance of the proposed SVIN algorithm.

Figure 4 shows a small selection of some sample maps and paths from the rock-world dataset. We split the rock-world data by maps into 90% training and 10% test. The reward network $f_R$ we use for rock-world consists of 2 convolutional layers with 1x1 kernels. The first has 16 output channels and relu nolinearity, the second has a single straight output channel. We use the Adam optimization algorithm in our gradient descent procedure (Kingma & Ba, 2014).

Figure 5 shows the result of our training results. The SVIN algorithm is able to train up to a very high level of performance on this dataset (90%). When using the hard action model the network plateaus around 70-75% accuracy.

It is also worth noting that the optimal upper performance bound is less than 100% and thus SVIN actual performance is more than 90% of the optimal solution. There are two reason for this: First, the rock-world environment often contains multiple paths of identical length, and, structurally, the

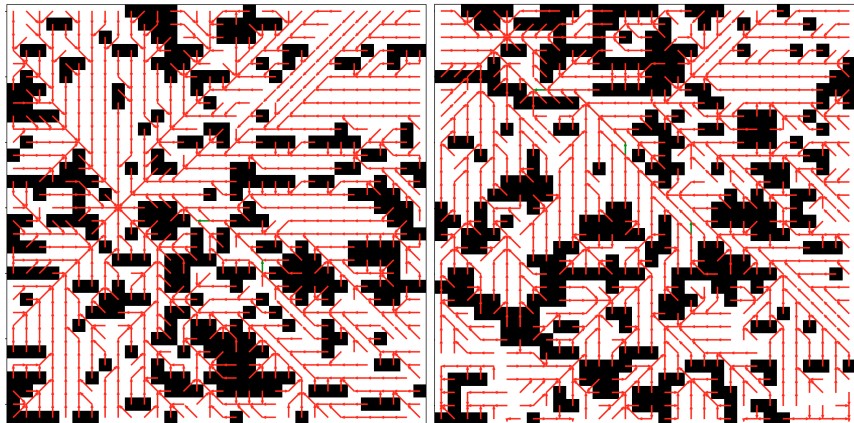

Figure 6: Vector fields show optimal policy planning results on our rock-world dataset. Learned policy actions are shown in red. A few green arrows show places where the optimal actions from the training data deviated from our learned (red) policy. Note how in the map on the right some regions do not show valid policies that converge to the goal (bottom-right and bottom-left corners). This is an artifact of fixing the number of iterations (i.e., $k$) in VI. When $k$ is too small, information about the goal location cannot propagate to the whole map.

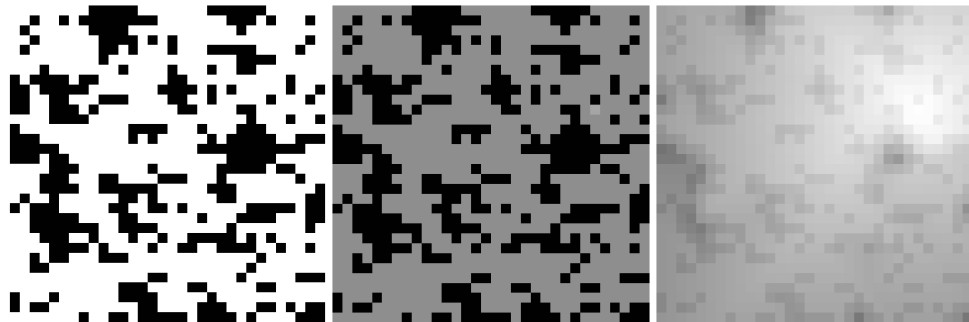

Figure 7: (left) An example rock-world obstacle map. (middle) The corresponding reward map after training. The goal position is the brightest square on the map. (right) Value map produced by the SVIN algorithm.

network is not capable of learning the same tie-breaking preferences. Additionally, the network is constrained by the choice of hyper-parameters. In particular, $k$ controls how far information can propagate through the network. Our rock-world network is trained with $k = 64$, which is longer than most paths, but still shorter than some paths in our 1000-path dataset.

Figure 6 further illustrate these concepts and potential failure modes. The learned policy is shown in red. A few green arrows show where the training path deviates from the learned policy. In both maps the goal position can be identified as the major point of convergence of the vector field. The left map shows some instances of deviations of identical length. The right map has some distant areas of the map (bottom-right and bottom-left corners) that show clearly incorrect behavior.

Figure 7 visualizes the behavior of a trained network. It shows the input rock map on the left, the result after passing through the reward network in the middle, and the output value map on the right. It can be seen that SVIN correctly identifies the goal position as the brightest square in the reward map (middle graph). Also, in the value map (right graph), we see how the obstacles appear to cast shadows through the space.

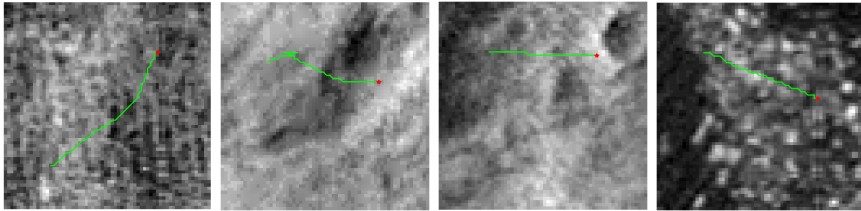

Figure 8: A selection of path segments from our MSL drive dataset.

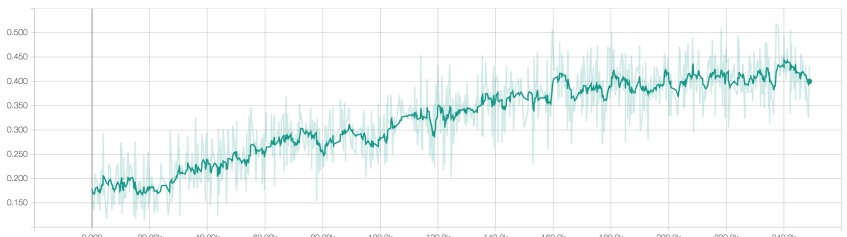

Figure 9: Accuracy training curve on the MSL dataset.

## 5.2 MSL DRIVE DATA

The MSL "Curiosity" rover has been on Mars for over 5 years now, and driven over 17km across the surface. We treat this driving history as the supervision behavior we wish to imitate by generating a dataset of segments of rover driving data. Overhead imagery is taken from the MSL 'basemap', which is a mosaic of imagery by the Mar Reconnaissance Orbiter HiRISE imager. This overhead map provides imagery of the surface with a resolution of 25cm per pixel. The imagery we use is all single channel (gray scale). Stereo interpolation of the HiRISE imagery also provides a digital elevation map (DEM) with a resolution of 1 meter per pixel. This will be incorporated into the planning in future work.

To create our dataset we break up the rover driving path into segments of similar length that all fit in identical size (64x64) tiles of the overhead imagery. To separate out a test set from that data we have separated a segment of the most recent part of the rover path that represents a little more than 20% of our samples. We expect systematic variations in terrain appearance in the test data, as the rover drives into new types of terrain. Also, this is aligned with the requirement for deploying SVIN on a rover that will navigate on new Mars terrains. Figure 8 shows a few example selection of path tiles from our dataset. The total dataset (train and test) is about 4700 tiles.

Selecting sections of the rover path is complicated due to the behavior of a science-gathering rover. Specifically, although the rover is driven with long-range strategic goals in mind, it is often diverted to examine interesting geological features or other science objectives. These behaviors will introduce noise in our dataset, unless we can identify the rover's sub-goals and use them as break points in dividing up the path. This is a subject of future work.

Due to the complexity of the data, for this dataset we use a slightly more complex reward network than the network we used for the rock-world. We use three convolutional layers, the first 11x11 with 16 channels, then 5x5 with 16 channels, and again the output layer is 1x1, 1 channel and linear. Figure 9 shows the training curve of the accuracy metric with the MSL dataset. The network achieves 45% accuracy on the noisy Mars navigation dataset. [1]

---

[1]We had to stop the training at the time of submission. The trend of the curve suggests performance will continue to improve with more training time. For the after-review version, we will be able to report the training results over a longer training time.

## 6 DISCUSSION AND FUTURE WORK

In this paper, we have developed a new variant of value iteration networks, referred to SVIN. The focus of SVIN is to solve motion planning problems via imitation learning. Our primary learning objective is to develop a network that can transform map data into a properly calibrated reward function. The SVIN approach gives us the differentiable planning algorithm and an informative gradient calculation necessary to accomplish this goal. We demonstrated this approach on a synthetic rock-world dataset, and showed some early results applying the technique to a much more challenging navigation problem with very noisy dataset for a rover navigation on Mars.

In our future work, we will be looking at ways to improve the performance on the Mars dataset. In particular, we will augment the Mars dataset with synthetic simulation data created from high-fidelity Mars terrain and Mars rover simulators. We will also experiment with using a pre-trained reward network trained on similar visual tasks the accelerate learning process. Second, we will look into selecting correct sub-goals on rovers paths in the Mars dataset. This will significantly reduce the noise in the dataset and enhance the learning results.

ACKNOWLEDGMENTS

Omitted for blind review.

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
