# OpenReview forum: "Soft Value Iteration Networks for Planetary Rover Path Planning"
_ICLR.cc/2018/Conference — Reject_

### Official Review · AnonReviewer1 · 2017-11-19
**Why behavior-cloning vs. imitation learning? Why SVIN/VIN vs traditional CNN or some other architecture?**

**Rating:** 3
**Confidence:** 5

**Review:**

Summary:

The Value-Iteration-Network (VIN) architecture is modified to have a softmax loss function at the end. This is termed SVIN. It is then applied in a behavior cloning manner to the task of rover path planning from start to goal from overhead imagery.

Simulation results on binary obstacle maps and using real-world Mars overhead orbiter maps are shown. On the simulation maps SVIN is shown to achieve 15-20% better lower training error than VIN.

One the Mars images it trains up to 45% training accuracy. (What was testing accuracy?)


Comments:

- Section 1.1: "Autonomous driving techniques also exist, though they lack the ability to do high level planning to choose paths that are advantageous for longer term navigation." --This is not true. See any of the numerous good systems described in literature. See the special editions of the Journal of Field Robotics on DARPA Urban Challenge and Desert Challenge or any of the special editions for the Learning Applied to Ground Robots (LAGR) program for excellent literature describing real-world autonomous ground vehicle systems. And specifically for the case of predicting good long-term trajectories from overhead imagery see: Sofman, B., Lin, E., Bagnell, J. A., Cole, J., Vandapel, N., & Stentz, A. (2006). Improving robot navigation through self‐supervised online learning. Journal of Field Robotics. (Papers related to this have been cited in this paper already).

- Section 4.1: "During training the output from the VI module is fed to an action selection function to compare those results against actions chosen in the training data.": What is the action selection function? Is it a local planner (e.g. receding-horizon model-predictive control)? Is it a global planner with access to full map to the goal (e.g. A* run all the way to the goal location assuming that during training the entire map is available)? Same question for Figure 2 where the 'expert action' block doesn't specify who is the expert here (computational or human).

- Section 2: "Imitation learning for navigation has been studied by other groups as well (Silver et al. 2010)": That particular paper is about using inverse optimal control (aka inverse reinforcement learning) and not imitation learning for first learning a good terrain cost function and then using it in a receding-horizon fashion. For imitation learning in navigation see "Learning Monocular Reactive UAV Control in Cluttered Natural Environments" by Ross et al. and relevant literature cited therein.

- My main concerns with the experiments is that they are not answering two main questions: 1. What is SVIN/VIN bringing to the table as a function approximator as opposed to using a more traditional but similar capacity CNN? 2. Why are the authors choosing to do essentially behavior cloning as opposed to imitation learning? It is well established (both theoretically and empirically) that imitation learning has mistake bounds which are linear in the time horizon while behavior cloning is quadratic. See Ross et al., "A Reduction of Imitation Learning and Structured Prediction to No-Regret Online Learning."

- Figure 6: Please mark the goal points. It is not obvious where it is from the field arrows.

- Figure 8: Are white regions high/low cost? It is not obvious from the pictures what is being avoided by the paths.

- What does 45% accuracy actually mean? Are the predicted paths still usable? No figures showing some qualitative good and bad examples are shown so hard to tell.

- On the rover overhead imagery if a simple A*/Dijkstra search algorithm was run from start to goal using the DEM as a heuristic cost map, how well will it do compared to SVIN?

---

> ### Author Response · Authors · 2018-01-06
> **Response to Reviewer 1**
>
> We thank the reviewer for their comments and suggestions.
>
> Re: testing accuracy: Testing accuracy in our experiments has been essentially identical to training accuracy, which we attribute to what amounts to a very strong prior from value iteration module.  We will include this comparison in the next revision.
>
> Re: Section 1.1: The reviewer is correct that there is substantial work in autonomous driving of cars and other terrestrial vehicles, however, this sentence is specifically referring to currently operational (certified for use on Mars) techniques for planetary rovers.
>
> Re: Section 4.1: The action selection block represent the process of turning Q values into an action.  The exact behavior here is described in section 4.2.  The expert action here comes from the trajectory demonstrations (training data), which, depending on the data source, may have been a human or computer.
>
> Regarding the reviewers main concerns:
> 1: A traditional CNN would not incorporate the planning structure of the MDP formulation of our problem.  The VIN/SVIN formulation produces a ‘value map’ as an intermediate result, which is the key data product for our application.
> 2: We dispute the reviewer’s assertion that our technique would not be described as imitation learning.

---

### Official Review · AnonReviewer2 · 2017-11-26
**Not enough novel contribution**

**Rating:** 3
**Confidence:** 3

**Review:**

To my understanding, the focus of the paper is to learn a reward function based on expert trajectories. To do so, they use a Value Iteration Module to make the planning step differentiable.
To improve training, they propose to replace the max-operator in value iteration by a softmax operator.

I believe the paper should be rejected. I will outline my major concerns below.

1. I have a difficult time identifying the (large enough) novel contribution. The algorithm uses automatic differentiation through an approximate value iteration (approximate because it is only performed for k iterations) to learn a reward function. To me this seems like a very straightforward case of inverse reinforcement learning (with which I admittedly am not too familiar). I think at least inverse RL should be mentioned and the relationship to it should be discussed in the paper (which it is currently not).
2. The experimental results are not convincing. In my opinion, the first experiment is too simple to showcase the algorithm. In particular, note that the number of total parameters learned are in the order of only 100 or less for the 2 layer reward network, depending on how many 'other layers' are used as input (and everything else I understand to be fixed). Furthermore, one input layer corresponds to a one hot encoding of the goal, which is also a correct output of the reward function. Consequently, the reward function must only learn the identity function (or a multiple thereof). This is further simplified by only using 1x1 convolutions. The second experiment doesn't have a baseline to compare the results against so it is hard to judge how well the algorithm performs.
3. I am not sure that there is enough similarity to Value Iteration Networks that it should be described as an extension thereof. As far as I understand it, the original Value Iteration Network consists of the Value-Iteration Module as well as several other modules. In the reviewed paper, it seems that only the Value Iteration Module is used. Furthermore, in the current paper, the transition kernels are not learned.


Smaller notes:
- One could include a background section briefly explaining Value Iteration Networks. Or leave out VINs completely as I'm not sure there is too much similarity. But I might be wrong.
- 1. Paragraph in "3. Preliminaries", last sentence: Albeit obvious, it should be included how $a_t$ and $s_t$ are drawn for $t>0$ and $t>1$ respectively
- In 4.2: Traninable => trainable
- In 4.3: As the algorithm is not actually tested on any mars rover, I wouldn't include that part in the "Algorithm" section. Maybe in the conclusion/outlook instead?
- In 4.4, second paragraph: stacked = added? I guess both would work but what would be the advantage of stacking, especially when the kernel is known and fixed (and I assume simply performs a discounted addition?)
- Please use a larger font in your plots
- Figure 6: While I like the idea of having a qualitative analysis of the results, it would be nice if red and green arrows would be easier to tell apart. The green ones are hard to find at the moment.

---

> ### Author Response · Authors · 2018-01-06
> **Response to Reviewer 2**
>
> We thank the reviewer for their comments and suggestions.
>
> Responding the reviewer's major concerns:
> 1. The reviewer is correct that inverse reinforcement learning and imitation learning are essentially interchangeable terms (though some may draw some minor distinctions in they way they are applied), though we disagree that the application is straightforward as most IL and IRL architectures do not include an explicit planning stage.
>
> 2. The gridworld environment is indeed quite simple, however, useful reward functions are deceptively complex.  If the relative costs and rewards of different states are not properly balanced they can produce counter-intuitive behavior, particularly when it takes many steps to reach a reward.
>
> 3. The reviewer is correct that Tamar et al's VIN paper included a number of other elements around the VI module.  We see the use of the VI module as one of the important innovations of that paper, and in our work we suggest modifications to the module as well as different surrounding components to serve a different application.

---

### Official Review · AnonReviewer3 · 2017-11-27
**Sound improvement over VIN, but missing baselines, citations**

**Rating:** 4
**Confidence:** 4

**Review:**

Summary:
The submission proposes a simple modification to the Value Iteration Networks (VIN) method of Tamar et al., basically consisting of assuming a stochastic policy and replacing the max-over-actions in value iteration with an expectation that weights actions proportional to their exponentiated Q-values. Since this change removes the main nondifferentiability of VINs, it is hypothesized that the resulting method will be easier to train than VINs, and experiments seem to support this hypothesis.

Pros:
+ The proposed modification to VIN is simple, well-motivated, and addresses the nondifferentiability of VIN
+ Experiments on synthetic data demonstrate a significant improvement over the standard VIN method

Cons:
+ Some important references are missing (e.g., MaxEnt IOC with deep-learned features)
+ Although intuitive, more detailed justification could be provided for replacing the max-over-actions with an exponentially-weighted average
+ No baselines are provided for the experiments with real data
+ All the experimental scenarios are fairly simple (2D grid-worlds with discrete actions, 1-channel input features)

The proposed method is simple, well-motivated, and addresses a real concern in VINs, which is their nondifferentiability. Although many of the nonlinearities used in CNNs for computer vision applications are nondifferentiable, the theoretical grounds for using these in conjunction with gradient-based optimization is obviously questionable. Despite this, they are widely used for such applications because of strong empirical results showing that such nonlinearities are beneficial in image-processing applications. However, it would be incorrect to assume that because such nonlinearities work for image processing, they are also beneficial in the context of unrolling value iteration.

Replacing the max-over-actions with an exponentially-weighted average is an intuitively well-motivated alternative because, as the authors note, it incorporates the values of suboptimal actions during the training procedure. We would therefore expect better or faster training, as the values of these suboptimal actions can be updated more frequently. The (admittedly limited) experiments bear out this hypothesis.

Perhaps the most significant downside of this work is that it fails to acknowledge prior work in the RL and IOC literature that result in similar  “smoothed” or “softmax" Bellman updates: in particular, MaxEnt IOC [A] and linearly-solvable MDPs [B] both fall in this category. Both of those papers clearly derive approximate Bellman equations from modified optimal control principles; although I believe this is also possible for the proposed update (Eq. 11), along the lines of the sentence after Eq. 11, this should be made more explicit/rigorous, and the result compared to [A,B].

Another important missing reference is [C], which learned cost maps with deep neural networks in a MaxEnt IOC framework. As far as I can tell, the application is identical to that of the present paper, and [C] may have some advantages: for instance, [C] features a principled, fully-differentiable training objective while also avoiding having to backprop through the inference procedure, as in VIN. Again, this raises the question of how the proposed method compares to MaxEnt IOC, both theoretically and experimentally.

The experiments are also a bit lacking in a few ways. First, a baseline is only provided for the experiments with synthetic data. Although that experiment shows a promising, significant advantage over VIN, the lack of baselines for the experiment with real data is disappointing. Furthermore, the setting for the experiments is fairly simple, consisting of a grid-world with 1-channel input features. The setting is simple enough that even shallow IOC methods (e.g., [D]) would probably perform well; however, the deep IOC methods of [C] is also applicable and should probably also be evaluated as a baseline.

In summary, although the method proposes an intuitively reasonable modification to VIN that seems to outperform it in limited experiments, the submission fails to acknowledge important related work (especially the MaxEnt IOC methods of [A,D]) that may have significant theoretical and practical advantages. Unfortunately, I believe the original VIN paper also failed to articulate the precise advantages of VIN over this prior work—which is not to say there are none, but it is clear that VINs applied to problems as simple as the one considered here have real competitors in prior work. Clarifying this connection, both theoretically and experimentally, would make this work much stronger and would be a valuable contribution to the literature.

[A] Ziebart, Brian D. Modeling purposeful adaptive behavior with the principle of maximum causal entropy. Carnegie Mellon University, 2010.
[B] Todorov, Emanuel. "Linearly-solvable Markov decision problems." Advances in neural information processing systems. 2007.
[C] Wulfmeier et al. Watch This: Scalable Cost-Function Learning for Path Planning in Urban Environments. IROS 2016
[D] Ratliff, Nathan D., David Silver, and J. Andrew Bagnell. "Learning to search: Functional gradient techniques for imitation learning." Autonomous Robots 27.1 (2009): 25-53.

---

> ### Author Response · Authors · 2018-01-06
> **Response to Reviewer 3**
>
> We thank the reviewer for their thorough comments, suggestions, and comparisons.
>
> The reviewer’s suggestions of additional baselines, particularly for the real data experiments, are well received and something we are currently working on.  In particular, the reviewer points out similarities to MaxEnt IOC techniques, and we believe these comparisons are quite interesting and worthy of further investigation, which we hope to incorporate in a future version of this paper.

---

### Decision · Program_Chairs · 2018-01-29
**ICLR 2018 Conference Acceptance Decision**

**Decision:**

Reject

**Comment:**

The authors have proposed a 'soft' version of VIN which is differentiable, where the cost function is trained by behavior cloning / imitation learning from expert/computer trajectories. The method is applied to a toy problem and to real historical data from mars rovers. The paper does not acknowledge nor compare against other methods, and the contribution is unclear, as is the justification for some of the aspects of the method.  Additionally it is difficult to interpret the relevance or significance of the results (45% correct).